# NOSE Augment: Fast and Effective Data Augmentation Without Searching

## Abstract

Data augmentation has been widely used for enhancing the diversity of training data and model generalization. Different from traditional handcrafted methods, recent research introduced automated search for optimal data augmentation policies and achieved state-of-the-art results on image classification tasks. However, these search-based implementations typically incur high computation cost and long search time because of large search spaces and complex searching algorithms. We revisited automated augmentation from alternate perspectives, such as increasing diversity and manipulating the overall usage of augmented data. In this paper, we present an augmentation method without policy searching called NOSE Augment (NO SEarch Augment). Our method completely skips policy searching; instead, it jointly applies multi-stage augmentation strategy and introduces more augmentation operations on top of a simple stochastic augmentation mechanism. With more augmentation operations, we boost the data diversity of stochastic augmentation; and with the phased complexity driven strategy, we ensure the whole training process converged smoothly to a good quality model. We conducted extensive experiments and showed that our method could match or surpass state-of-the-art results provided by search-based methods in terms of accuracies. Without the need for policy search, our method is much more efficient than the existing AutoAugment series of methods. Besides image classification, we also examine the general validity of our proposed method by applying our method to Face Recognition and Text Detection of the Optical Character Recognition (OCR) problems. The results establish our proposed method as a fast and competitive data augmentation strategy that can be used across various CV tasks.

## 1 Introduction

Data is an essential and dominant factor for learning AI models, especially in deep learning era where deep neural networks normally require large data volume for training. Data augmentation techniques artificially create new samples to increase the diversity of training data and in turn the generalization of AI models. For example, different image transformation operations, such as rotation, flip, shear etc., have been used to generate variations on original image samples in image classification and other computer vision tasks. More intricate augmentation operations have also been implemented, such as Cutout (Devries & Taylor, 2017), Mixup (Zhang et al., 2018), Cutmix (Yun et al., 2019), Sample Pairing (Inoue, 2018), and so on. How to formulate effective augmentation strategies with these basic augmentation methods becomes the crucial factor to the success of data augmentation.

Recent works (Cubuk et al., 2019; Lim et al., 2019; Ho et al., 2019) introduced automated searching or optimization techniques in augmentation policy search. The common assumption of these methods is: a selected subset of better-fit augmentation policies will produce more relevant augmented data which will in turn result in a better trained model. Here the augmentation policy is defined by an ordered sequence of augmentation operations, such as image transformations, parameterized with probability and magnitude. Though these methods achieved state-of-the-art accuracies on image classification tasks, they lead to high computational cost in general, due to large search space and extra training steps. More importantly, it is worth exploring whether it is really necessary to find the best-fit subset of policies with specific parameter values of probability and magnitude. RandAugment (Cubuk et al., 2020) has started to simplify the parameters and scale down the search

space defined by AutoAugment Cubuk et al. (2019), but their method still relied on grid search for iterative optimization of the simplified parameters.

Our method aims to fully avoid policy search and cost, meanwhile to maintain or improve model performance in terms of both accuracy and training efficiency. Our work showed that by applying simple stochastic augmentation policies with the same sampling space and other settings of training, we could obtain equal or very close performance with search-based augmentation methods. Another advantage of stochastic policies is that adding more operations in the pool does not bring additional cost; while in search-base methods, more operations in the pool causes exponential increase of the search space. Therefore, the second part of our method is to add more operations to the pool to bring more data diversity. In practice, we introduced a new category of operations such as mixup and cutmix into the operation pool. Furthermore, we tackled automated augmentation from overall data usage point of view, in contrast to data creation point of view accentuated by policy-search based methods. Inspired by the idea of Curriculum Learning (CL) (Bengio et al., 2009), which presents training samples in an increasing order of difficulties, our method defines various complexity levels of augmentation strategies and applies them with orders on phased training stages. To avoid the confounding overfitting problem of original Curriculum Learning in practice, our method applies the inverted order of Curriculum Learning, which presents the hardest augmentation strategies from the beginning and gradually decreases the complexity levels.

In general, our augmentation method replaces policy search with stochastic policy generation, upon which it introduces more operations for better diversity and phased augmentation strategy with decreasing complexities for a smooth learning convergence, and as an integral solution it achieves better results. Figure 1 describes our method and the difference compared to search-based methods.

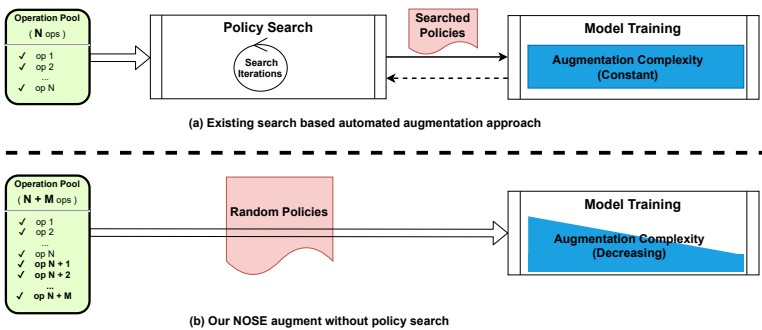

Figure 1: No Search (NOSE) Augment vs Search-based Augment

The main contributions of this paper can be summarized as follows:

1. We present a no-search (NOSE) augmentation method as an alternative of computation-intensive search-based auto-augment methods. By jointly applying phased augmentation strategy and introducing more augmentation operations on top of a simple stochastic augmentation mechanism, NOSE augment achieves state-of-the-art (SOTA) accuracies on CIFAR 10, CIFAR 100 (Krizhevsky, 2009) and close-to-SOTA results on other benchmark datasets. Our ablation study demonstrates that all the components of our methods should be combined together to achieve the best performance.

2. We demonstrate that a stochastic-based augmentation approach can obtain accuracies comparable to those of search-based methods while achieving overwhelming advantage on overall augmentation and training efficiency as the searching phase is completely avoided.

3. Besides image classification, we also applied NOSE augment on face recognition and text detection (OCR) tasks, and obtained competitive or better results in comparison with search-based methods. This further proves the advantage and generality of NOSE augment.

## 2 RELATED WORKS

Our work is most closely related to RandAugment (Cubuk et al., 2020). A key idea in RandAugment is to avoid a separate search phase on proxy datasets but chose instead to reduce the search space by decreasing the number of parameters. For instance, the probability for each operator is set to a constant uniform probability. The two sets of data augmentation parameters to be searched are N, the number of transformations for a given image and M, the discrete operator magnitude. These two parameters are regarded as hyperparameters and tuned using grid search. Our method is similar to RandAugment in terms of the operators used and the setting of equal probability for each operator. The most contrasting difference between RandAugment and our method is that RandAugment is still a search-based method with search cost while our method skips policy search completely and tackles the problem by introducing new augmentation operations and further new strategies built upon stochastic policies.

Our work stands diametrically opposite to the AutoAugment series of methods (Cubuk et al., 2019; Lim et al., 2019; Zhang et al., 2020), in which the best data augmentation policy is derived through a search. AutoAugment (Cubuk et al., 2019) uses a RNN controller to propose an augmentation strategy. Based on this proposed augmentation strategy, the model is trained and use the resulting validation accuracy to update the RNN controller. Although AutoAugment achieves good results on the standard benchmark datasets, the search cost is prohibitively huge, for instance, 5000 GPU (NVIDIA Tesla P100) hours on the CIFAR-10 dataset and Pyramid-Net+ShakeDrop model.

Fast AutoAugment (Fast AA) (Lim et al., 2019) ameliorates AutoAugment's huge search cost with a three pronged approach. Smaller datasets are used for policy search. The idea of density matching is proposed to avoid having to re-train the model for assessing the validation accuracy for each proposed policy. The use of Bayesian Optimization based method (Bergstra et al., 2011) also helps to converge to an effective augmentation policy quickly. Adversarial Augment (Adv AA) (Zhang et al., 2020) reduces the huge computational cost of AutoAugment through the adversarial policy framework, which generates data samples that maximizes the training loss of the target network. PBA (Ho et al., 2019) introduces the idea of non-stationary policy schedules instead of the fixed augmentation policy proposed in AutoAugment. The non-stationary policy schedules refers to how the policy evolves with the training epochs.

Our main differences with aforementioned state-of-the-art methods lie in our stochastic policy described in section 3.1 and the multi-stage complexity driven augmentation policy outlined in section 3.3. We use the same set of operators and settings as these related works.

## 3 METHOD

We describe the three key components of our proposed method in this section. In section 3.1, we put forward a stochastic-based method as opposed to the search-based paradigm advocated by many of the state-of-the-art works described in section 2. Due to the counter-intuitiveness (but effective) nature of this stochastic-based method, we first provide insights and motivations into stochastic augmentation policies in section 3.1. In section 3.2, we capitalize on our proposed stochastic-based method by proposing additional operators that will further enrich the data augmentation diversity, since there is no extra search cost for adding new operators with a no-search method. In 3.3, we propose a multi-stage complexity driven policy that helps to resolve the tension between augmented data diversity and data distribution fidelity. Our ablation study demonstrates that the performance of our method is not a simple incremental benefit with each individual component, but rather, the three components need to work together to achieve a competitive performance with no policy search cost.

### 3.1 STOCHASTIC POLICY - SKIPPING POLICY SEARCH

The first component of our method removes policy searching completely and applies a stochastic augmentation policy with randomly selected operations and magnitudes. Unlike the search-based methods, once an operation is selected, it is used with 100% probability. Our stochastic method follows the same policy definition as existing search-based methods. Specifically, one augmentation policy has 5 sub-policies; each sub-policy consists of 2 augmentation operations. The base operation pool includes the following 15 operations: ShearX/Y, TranslateX/Y, Rotate, AutoContrast, Invert,

Equalize, Solarize, Posterize, Contrast, Color, Brightness, Sharpness and Cutout. Each operation has 11 uniformly discretized magnitudes which is randomly selected upon each use.

We provide an abstract analysis for stochastic policy through a data deficiency complement point of view, in which a relatively slower accuracy increase is expected for stochastic method in early stage of training. We refer to the time period required by stochastic approach to accumulate enough amount of data in the deficient dimensions as the Stochastic Accumulation Stage (SAS). However, as training carries on, the amount of data of deficient dimension in the random approach may gradually get close to or even go beyond the one in search-based methods. When enough number of epochs is reached, the performance of random policy may match or even overtake search-based methods. More detailed intuitions are provided in section A.1 of the appendix.

Note that we are not claiming that this stochastic policy outperforms the search-based augmentation policies, but rather, this stochastic policy provides a good foundation for us to build upon; it allows us to further incorporate additional operators and a multi-stage complexity augmentation strategy with very little or no extra cost compared to search-based methods.

## 3.2 INTRODUCING ADDITIONAL AUGMENTATION OPERATIONS

In our method, we introduce mix-based operations such as mixup, cutmix, and augmix (Hendrycks et al., 2020) in addition to the randomly generated augmentation policies. RandAugment showed certain operation (e.g. posterize) brought consistent negative effect to augmentation result regardless of the number of operations in the pool; while some (e.g. rotate) had consistent positive effect. We also observed some operations might harm the augmentation performance when directly applied on stochastic method; however, an interesting finding is that the negative influence could be weakened or even turned back to positive when these operations were used together with our complexity driven strategy, which is explained in next section.

Note that these mix-based operators can also be introduced to the search-based methods. The main difference is that there is no extra overhead or cost as far as our method is concern; whereas these additional operators will incur even larger cost for an already expensive search process.

## 3.3 AUGMENTATION WITH MULTI-STAGE COMPLEXITY DRIVEN STRATEGY

Curriculum learning (CL) puts forward the view that learning progressively harder tasks may improve training performance. Drawing inspiration from their work, we manipulate the overall data complexity in different training stages instead of controlling the creation of static augmentation policies. The data complexity here refers to the distortion produced by the data augmentation operators and the result of successive applications of these operators. In our work, these augmentation operators are first grouped into three categories: 1) baseline operators such as flip, random crop, and cutout which are frequently used as fundamental augmentations for image-related tasks; 2) mix-based operations such as mixup, cutmix, and augmix; and 3) transformation-based operators such as rotate, shear, sharpness etc., which have been used in experiments of related works (Cubuk et al., 2019; 2020; Lim et al., 2019; Zhang et al., 2020). Our method then divides the complexity of augmentation into multiple levels that map to various combinations of the above three categories, and apply them in different training stages. Starting from the simplest, three complexity levels are defined as follow. First, baseline augmentation (BaseAug), exactly the same baseline augmentation used in AutoAugment; depending on the specific datasets, BaseAug may consist of flip, random crop, or cutout operations in the baseline category. Second, advanced augmentation (AdAug), which introduces the category of mix-based operations upon BaseAug. Third, super augmentation (SupAug), which additionally applies 15 transformation operators upon AdAug. Except for baseline augmentations which are applied with 100% probability, mix-based and transformation augmentations are selected in a stochastic manner when the corresponding augmentation category is applied in a certain augmentation stage.

For the overall augmentation strategy, we first tried applying augmentation complexity in ascending order while the training proceeded, same as Curriculum Learning. However, the result was not ideal and we noticed apparent trend of over-fitting in training. Our interpretation of the phenomenon is that the over-fitting is due to insufficient diversity of training data. When this insufficiency takes place in early training stages of a deep neural network (DNN), it is prone to over-fit because the

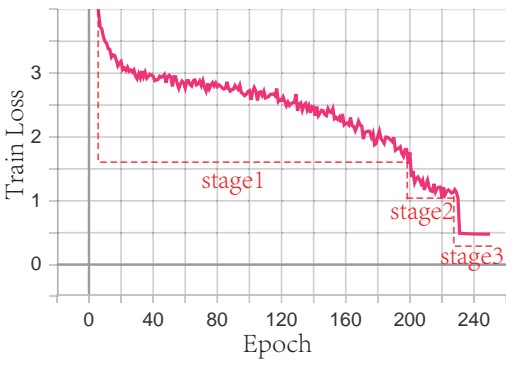
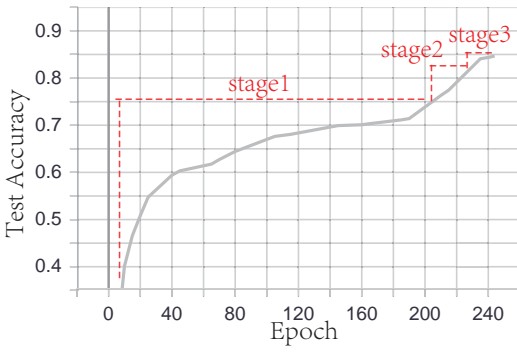

(a) Training loss with 3-phase augmentation        (b) Test accuracy (best) with 3-phase augmentation

Figure 2: Phased Augmentation

relatively smaller amount of data with limited diversity cannot support learning DNNs with large sizes and complex structures.

Above-mentioned observation led us to apply the inverted CL order of augmentation complexities, which trains the network with sufficiently diversified data in the early stage and gradually adapts the network to the original data distribution without augmentation. Specifically, our method removes one augmentation category going from one stage to the next e.g. the transformation category is removed going from stage 1 to stage 2. Note that our method does not tune or search the optimal epoch allocation during this process, it has a common setting of epoch allocation for a three-phase augmentation strategy. Here the first stage is set to have the highest complexity and the majority of the epochs, it is necessary to maintain the high complexity and enough epochs to obtain sufficient data diversity and prevent overfitting. Details of phase definition are further described in Section 4, and our experiments verified our hypothesis and confirmed the effectiveness of this method. Figure 2 shows example curves of training loss and test accuracy with a three-stage augmentation strategy obtained in practice; jumps can be seen on boundaries of adjacent stages.

We can also explain the need for data diversity vs. fidelity to the original data distribution in terms of the data augmentation theoretical foundation constructed in (Dao et al., 2019). In this work, data augmentation is regarded as a perturbation about the original data. Thus, using a Taylor expansion about the original data, the first order effect of data augmentation is given in equation 1. From equation 1, we can therefore understand data augmentation as *feature averaging* over the data augmentation operators. This is precisely the reason why towards the end of training, we want to keep only data augmentation operators that introduce minimum distortion, thus allowing convergence to a model that can best learn the original data distribution, instead of the augmented distribution.

$$g(w) \approx \hat{g}(w) \coloneqq \frac{1}{n} \sum_{i=1}^{n} l(w^T \, \mathbb{E}_{t_i \sim T(x_i)} \left[ \phi(t_i) \right]; y_i) \tag{1}$$

where $g(.)$ is the loss function after data augmentation, $l(.)$ is the original loss function, $w$ are the weights to be learned, $t_i$ is the data augmentation operator sampled from the operator pool $T$, $\hat{g}(.)$ is the expansion at any data point not dependent on $t_i$, $\phi$ is the feature map and $y_i$ is the label.

The second order expansion is given in equation 2. The squared term in equation 2 gives a clear indication of the role of data augmentation towards *variance regularization*, helping to prevent overfitting. This gives strong justification to why we chose higher complexity data augmentation operators in the early part of the training, which serves to provide strong regularization effect for preventing choosing $w$ that will lead to over-fitting in the latter part of training.

$$g(w) \approx \hat{g}(w) + \frac{1}{2n} \sum_{i=1}^{n} \mathbb{E}_{t_i \sim T(x_i)} \left[ w^T \left( \phi(t_i) - \psi(t_i) \right)^2 l''(\zeta_i(w^T \phi(t_i)); y_i) \right] \tag{2}$$

where $\psi(x_i) = \mathbb{E}_{t_i \sim T(x_i)}[\phi(t_i)]$ is the expectation of the original feature map over the data augmentation operators and $l''(.)$ is the remainder function from Taylor's theorem. For more mathematical details, we highly recommend the readers to refer to (Dao et al., 2019).

Each of the above-mentioned three sub-methods may benefit augmentation individually; however the best result is achieved by combining them together as one solution, NOSE augment, whose performance is demonstrated with our experiments presented as follows.

## 4    EXPERIMENTS AND RESULTS

In this section, we first evaluate the performance of NOSE augment in terms of accuracy and efficiency on CIFAR-10/CIFAR-100, Reduced CIFAR-10 which includes 4K training samples (randomly chosen), SVHN (Netzer et al., 2011) and ImageNet (Deng et al., 2009). NOSE achieves state-of-the-art results on many datasets or models, and competitive ones on the rest. We compare our method with our baseline and existing search-based methods described in section 2, namely AutoAugment, Fast AA, RandAugment, Adv AA and demonstrates both the efficiency and effectiveness of our method. We then present our extensive experiments on sub-methods of NOSE and ablation studies. Besides image classification tasks, we further conduct experiments on Face Recognition and Text Detection tasks, which further demonstrate the transferability of our method across domains.

### 4.1    OVERALL EXPERIMENT SETUP

As presented in Section 3, NOSE relies on three sub-methods including stochastic policies, extra operations, and staged augmentation strategy. One stochastic policy is composed of 2 operations, which are randomly selected from the base operation pool as described in Section 3.1. Each operation randomly takes one of the 11 discretized magnitudes which uniformly space between 0 and 1 when it is used. Once the operation is selected, it is applied with 100% probability. Upon stochastic policies, mix-based operations including Mixup, Cutmix, and Augmix are added into the operation pool. Augmix is applied on individual images when selected, same as existing operations in the pool; while Mixup and Cutmix are randomly applied on individual batches due to the inherent pairing logic of these operations. We then applied 3 stages of augmentation along training, as mentioned in Section 3.3. For the first stage, we augment the training data with the highest complexity (SupAug); followed by the 2nd stage with slightly lower complexity (AdAug); and finally the third stage with the lowest complexity (BaseAug). We examined NOSE with extensive experiments on multiple datasets listed below. Please refer to section A.3 for the training details. We follow AutoAugment and successive related works for baseline settings, we provide our reproduced baseline results together with those reported by AutoAugment in Table 1.

### 4.2    EXPERIMENTS ON ACCURACY

As shown in Table 1, our NOSE augment is able to achieve competitive accuracies on most datasets and models compared to other SOTA works. In particular, NOSE augment obtains 0.56% SOTA accuracy improvement with Wide-ResNet-40-2 on CIFAR-10, while on CIFAR-100 it obtains 0.87% SOTA increment with Wide-ResNet-40-2 and 0.44% SOTA increment with Wide-ResNet-28-10. Moreover, for all models trained on reduced CIFAR-10, NOSE augment achieves the highest scores when compared to AutoAugment, PBA and RandAugment. On ImageNet, NOSE achieve 0.65% accuracy increment compared to most search-based methods (AutoAugment, Fast AA, RandAugment) except Adv AA.

### 4.3    EXPERIMENTS ON EFFICIENCY

This experiment examines the efficiency of NOSE augment and compares the results with search-based methods. The computation cost in this context has two major components: searching (or tuning) cost and training cost. For the searching part, the cost of a search-based method is normally proportional to some factors such as model size, dataset size, number of the operations in augmentation pool etc. For example, Fast AA's policy-search cost is 3.5 GPU-hours for wresnet 40-2 with reduced Cifar-10, but 780 GPU-hours for Pyramid-Net+ShakeDrop on full Cifar-10 dataset. Table 2

Table 1: Top1 test accuracy(%) on various datasets and models. Our baseline follows the same baseline settings of AutoAugment; the first two columns show our reproduced baseline results and the ones reported by AutoAugment respectively.

| DataSet | Model | Baseline(our) | Baseline(AA) | Cutout | AA | PBA | FastAA | RA | AdvAA | Ours |
|---|---|---|---|---|---|---|---|---|---|---|
| | Wide-ResNet-40-2 | 94.52 | 94.70 | 95.90 | 96.30 | - | 96.30 | - | - | **96.86** |
| | Wide-ResNet-28-10 | 95.43 | 96.10 | 96.90 | 97.40 | 97.40 | 97.30 | 97.30 | **98.10** | 97.97 |
| CIFAR-10 | Shake-Shake(26 2x32d) | 95.78 | 96.40 | 97.00 | 97.50 | 97.50 | 97.50 | - | **97.64** | 97.54 |
| | Shake-Shake(26 2x96d) | 96.65 | 97.10 | 97.40 | 98.00 | 98.00 | 98.00 | 98.00 | 98.15 | **98.3** |
| | Shake-Shake(26 2x112d) | 96.68 | 97.20 | 97.40 | 98.10 | 98.00 | 98.10 | - | 98.22 | **98.31** |
| | PyramidNet+ShakeDrop | - | 97.30 | 97.70 | 98.50 | 98.50 | 98.30 | 98.50 | **98.64** | 98.57 |
| | Wide-ResNet-28-2 | 80.10 | - | 81.94 | 85.60 | - | - | 85.30 | - | **87.38** |
| Reduced CIFAR-10 | Wide-ResNet-28-10 | 81.17 | 81.20 | 83.50 | 87.70 | 87.18 | - | 86.80 | - | **89.18** |
| | Shake-Shake(26 2x96d) | 80.26 | 82.90 | 86.60 | **89.98** | 89.25 | - | - | - | **89.98** |
| | Wide-ResNet-40-2 | 74.18 | 74.00 | 74.80 | 79.30 | - | 79.40 | - | - | **80.27** |
| CIFAR-100 | Wide-ResNet-28-10 | 80.79 | 81.20 | 81.60 | 82.90 | 83.30 | 82.70 | 83.30 | 84.51 | **84.95** |
| | Shake-Shake(26 2x96d) | 79.77 | 82.90 | 84.00 | 85.70 | 84.70 | 85.40 | - | **85.90** | 85.31 |
| SVHN (core set) | Wide-ResNet-28-10 | 96.57 | 96.90 | - | 98.10 | - | - | **98.30** | - | 98.06 |
| IMAGENET | ResNet50 | - | 76.30 | - | 77.60 | - | 77.60 | 77.60 | **79.40** | 78.52 |

shows the searching costs of different augmentation methods. Here, Adv AA claimed its search cost close to 0 as it does not have a separate policy search phase, but its training cost is much higher than our method, which is further explained as follow. For the training part, NOSE augment has similar cost with the majority of search-based methods like AutoAugment and Fast AA, which process the training samples in one round. Suppose the training cost is a constant value $\mathcal{C}$, which is affected by the dataset size and the number of total training epochs. For Adv AA, the training cost is at least 8 times $\mathcal{C}$, because it augments more data in one epoch (number of batches is 8 times compared to NOSE, AutoAugment, and Fast AA). Rand Augment has significantly reduced the search space to 10x10, but it still relies on grid search with training and has a relatively higher training cost than NOSE augment. In general, our method has the overwhelming advantage in terms of the overall efficiency as it bypasses policy search completely and does not increase training cost with any tuning or optimization logic.

Table 2: Efficiency on various dataset. Unit: GPU hour; hardware: Fast AA - Tesla V100, AutoAugment - Tesla P100, PBA - Titan XP. Dashes indicate unavailable figures; despite the unavailability, the advantage of our method in terms of search cost is apparent.

| | Search Cost | | | |
|---|---|---|---|---|
| Method | Search Space | Reduced CIFAR-10 (Wide-ResNet-40-2) | CIFAR-10 (PyramidNet) | Reduced ImageNet (Wide-ResNet-40-2) |
| AA | $10^{32}$ | 5000 | - | 15000 |
| FastAA | $10^{32}$ | 3.5 | 780 | 450 |
| PBA | $10^{61}$ | 5 | - | - |
| Our | 0 | 0 | 0 | 0 |

## 4.4 EXPERIMENTS ON STOCHASTIC POLICIES

This experiment presents potential factors that could influence the performance of stochastic policies. The factors we explore are: 1) number of operations per sub-policy; 2) number of total operations in the augmentation operation pool. Experiment results of the first factor showed that when the number of operations in a sub-policy is larger than 3, the downgrade in performance is clear; our interpretation is that overlapped operations beyond certain threshold may cause downgrade in image quality which will in turn affect the augmentation performance negatively. Similar to what RandAugment found in their experiment, our experiment results for the second factor revealed that in general more operations in the candidate pool can benefit augmentation as it brings more data diversity, which inspire us to introduce additional operations in our method. For further details, please refer to the section A.4.

## 4.5 EXPERIMENTS ON MULTI-STAGE AUGMENTATION STRATEGY

As described in section 3.3, our data augmentation consists of three stages: stage 1 (SupAug), stage 2 (AdAug), stage 3 (BaseAug). We first look at the effect of two stages vs. three stages. The two stage strategy is constructed by choosing two of the three stages. From Table 3a, we see that while "stage 1 + stage 3" has the best two stage performance, our three stage augmentation strategy still offers the best overall performance. The importance of having the minimal distortion stage 3 is also emphasized from the observation of a sharp accuracy drop for the "stage 1 + stage 2" combination.

Next we look at the epoch allocation for each stage. We vary the number of epochs for each stage on the CIFAR-100 dataset. From Table 3b, we derive an overall principle for epoch allocation: stage 1 needs to have the majority of the epoch, about 85% from our experiment; and stage 2 and 3 are set as 10% and 5% respectively. This is used as common strategy and one-time setting for all cases; no tuning or searching is used in our method for epoch allocation. The total number of epochs for all experiments in this section is set to 200. Figure 6 in the appendix shows the training loss profile for these different epoch allocation.

We also justify the need for an inverted CL by comparing with the standard CL, where the augmentation stages go from simple to complex. As shown in Table 4, there is a noticeable performance drop associated with standard CL (simple to complex), compared to our proposed inverted CL.

Table 3: Multi-stage augmentation study on 2 vs. 3 stages and epoch allocation for each stage.

|  | Stage 1,2 | Stage 1,3 | Stage 2,3 | Stage 1,2,3 |
| --- | --- | --- | --- | --- |
| CIFAR-10 | 93.01 | 97.44 | 96.94 | **97.75** |
| CIFAR-100 | 73.89 | 79.33 | 77.78 | **79.89** |

(a) A comparison of two stage vs. our proposed three stage augmentation strategy accuracy on the CIFAR-10 and CIFAR-100, using Wide-ResNet-40-2 and Wide-ResNet-28-10 respectively

| Stage 1 | Stage 2 | Stage 3 | Accuracy |
| --- | --- | --- | --- |
| 50 | 100 | 50 | 78.37 |
| 50 | 50 | 100 | 77.92 |
| 70 | 70 | 60 | 77.98 |
| 170 | 20 | 10 | **79.89** |

(b) Accuracy on the CIFAR-100 dataset by varying epoch allocation for each stage

Table 4: Low to high complexity (increasing image distortion).

| DataSet | Model | Simple To Complex | Ours |
| --- | --- | --- | --- |
| CIFAR-10 | Wide-ResNet-28-10 | 92.76 | **97.75** |
| CIFAR-100 | Wide-ResNet-40-2 | 72.6 | **79.89** |

## 4.6 EXPERIMENTS ON FACE RECOGNITION

Face recognition system is trained using the CASIAWebFace (Yi et al., 2014) dataset. CASIA dataset contains 10575 subjects and around 500k face images of the subjects. We used the test sets of LFW (Huang et al., 2008), AgeDB-30 (Moschoglou et al., 2017) and CFP-FP (S. Sengupta, 2016). The training details of our experiments for face recognition task are given in section A.6. As shown in Table 5a our model improved over the baseline result and showed similar accuracy results compared to Fast AA; while our method achieves much better training efficiency.

## 4.7 EXPERIMENTS ON TEXT DETECTION

We conducted experiments on ICDAR 2017 MLT dataset (Nayef et al., 2017) to test the generality of our method for text detection tasks. We trained our implementation of EAST (Zhou et al., 2017) model with 3 different augmentation strategies. The training details of our experiments for text detection task are given in section A.7. We performed several independent runs and averaged the results that are shown in Table 5b. In addition, we also included the results of our customized metric for each trained model to show its consistency. Further details about the customized metric which we call it as E-Score are given in section A.7. According to the final results, our approach showed

Table 5: Face recognition and text detection results

|  | Default | Fast AA | Ours |
|---|---|---|---|
| LFW | 99.20 | **99.27** | 99.25 |
| AgeDB-30 | 91.82 | 91.88 | **91.92** |
| CFP-FP | 94.31 | **95.53** | 95.17 |

(a) Face recognition accuracy on the standard benchmark dataset

|  | Default | Fast AA | Ours |
|---|---|---|---|
| F1 | 50.36 | 50.51 | **54.27** |
| Precision | 44.64 | 44.1 | **50.96** |
| Recall | 57.76 | **59.1** | 58.04 |
| Avg. Precision | 46.58 | 48.29 | **48.35** |
| E-Score | 1.44 | 1.46 | **1.47** |

(b) Text detection results on the ICDAR MLT 2017 dataset. Note that E-Score is our customized metric.

better performance than other approaches, with the model trained using our approach gains ∼4% advantage on the final F1 score compared to the model trained by default training approach with using baseline augmentations and the model trained by using the augmentation policies found via Fast AA.

## 4.8 ABLATION EXPERIMENTS

We design ablation studies to explore the benefits brought by individual components of our proposed method. The first control experiment evaluates stochastic policy ("Stoch. + Base"), in which transformation augmentations are applied randomly on top of baseline augmentations while mix-based operations and stage concept are removed. The second control experiment evaluates phased augmentation strategy without mix-based augmentations ("Stoch. + 2-stage"), where the first stage consists of stochastic transformations on top of baseline augmentations and the second stage with only the baseline augmentations. Note that 2-stage but not 3-stage is used here because removing mix-based augmentations removes the corresponding phase, stochastic augmentations have to be kept otherwise there will be only one stage left. The third control experiment evaluates the mix-based operations on top of stochastic transformations without phased augmentation strategy ("Stoch. + mix-based"). As we can see from Table 6, using partially selected components of our proposed method results in apparent accuracy drops in all cases, thus validating that our method as a whole is greater than the sum of its parts.

Table 6: Ablation study result. In "Stoch. + 2-stage", the first stage consists of stochastic transformations on top of baseline augmentations and only the baseline augmentations for the second stage.

|  | Model | Stoch.+ Base | Stoch.+ 2-stage | Stoch.+ mix-based | Ours |
|---|---|---|---|---|---|
| CIFAR-10 | Wide-ResNet-40-2 | 96.28 | 96.3 | 92.87 | **97.94** |
|  | Wide-ResNet-28-10 | 97.43 | 97.51 | 96.49 | **97.75** |
| CIFAR-100 | Wide-ResNet-40-2 | 79.12 | 79.34 | 72.56 | **79.89** |
|  | Wide-ResNet-28-10 | 82.61 | 83.11 | 79.49 | **85.17** |

## 5 CONCLUSION AND DISCUSSION

In this paper, we present an automated augmentation method without policy search. Upon stochastic policies, our method introduces more augmentation methods without extra search cost; by further applying a multi-stage complexity driven augmentation strategy, our method achieved state-of-the-art accuracies for image classification task on most datasets and absolute advantage on efficiency as policy search is skipped. We also apply this method on face recognition and text detection tasks, thereby demonstrating the generality of our method. Despite the strong performance of our proposed method, we are not denigrating the search-based methods and downplaying their values. On the contrary, we believe search-based methods can be further improved with revised or different search spaces. One of our future works is applying automated search techniques on our method for new augmentation operation selection and staged augmentation strategy optimization.

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

## A APPENDIX

### A.1 STOCHASTIC POLICY INSIGHT

We provide a motivation for stochastic policy through a data deficiency complement point of view. We think of the training data as a pool of knowledge or features with multiple dimensions. The columns in the histogram indicate the volume of data with this knowledge dimension. With searched static policies of AutoAugment methods (as shown in Figure 4a), the buckets of the most deficient data dimensions (e.g. dimension 2 in the figure) can be filled fast once training starts and catch up with other dimensions in a relatively short period. In contrast, data growing with random policies are balanced amongst all dimensions in a larger sampling scope (Figure 4b), so data accumulation on the dimensions of deficient data may be slower than search-based methods. Hence, in early stage of training, a relatively slower accuracy increase is expected. We refer the time period required by stochastic approach to accumulate enough amount of data in the deficient dimensions as Stochastic Accumulation Stage (SAS).

However, the relative deficiency are changing over time along training (as shown in Figure 4a); with static policies searched by algorithms such as AutoAugment and Fast AA, the constant increase on the focused dimensions may not fit the dynamic state of training and the data augmentation on the latest deficient dimensions slows down. Therefore, as training time goes, the data amount of deficient dimension in random approach may gradually get close to or even go beyond the one in search-based methods. When enough number of epochs is reached, the performance of random policy may match or overtake search-based methods. This prediction is verified by comparing practical training results between stochastic method and Fast AA, as shown in Figure 3.

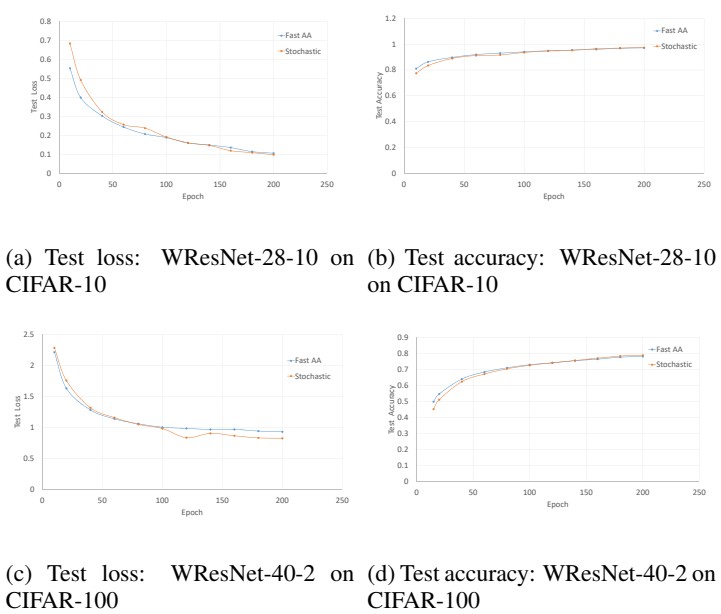

(a) Test loss: WResNet-28-10 on CIFAR-10

(b) Test accuracy: WResNet-28-10 on CIFAR-10

(c) Test loss: WResNet-40-2 on CIFAR-100

(d) Test accuracy: WResNet-40-2 on CIFAR-100

Figure 3: Training with Augmentation: Stochastic VS Fast AA

Our hypothesis is confirmed with experiment results on multiple datasets, as shown in Figure 4. As we can see in the figure, the SAS periods of stochastic method are shorter than the total epochs spent on achieving optimal results in search-based methods, thus the slower accumulation of deficient data with random policies can be ignored while the benefit is it completely skips the expensive policy searching. We have to mention that succeeding research in Adversarial AutoAugment exploited adversarial network for generating dynamic policies which to some extent made up for the disadvantages of static policies. But the time and cost of policy searching were still existing and considerable, compared to the stochastic approach without any searching.

First of all, our method removes policy searching completely and applies stochastic augmentation policies with randomly selected operations and magnitudes. Once an operation is selected, it is used with 100% probability. Our stochastic method follows the same policy definition as existing search-based methods. Specifically, one augmentation policy has 5 sub-policies; each sub-policy consists of 2 augmentation operations. The operation pool includes the following 15 operations: ShearX/Y, TranslateX/Y, Rotate, AutoContrast, Invert, Equalize, Solarize, Posterize, Contrast, Color, Brightness, Sharpness and Cutout. Each operation has 11 uniformly discretized magnitudes which is randomly selected upon each use.

Why can a simple stochastic mechanism match a searched one? Intuitively, learning result based on the superset data generated by random policies should not be worse than the subset data generated by searched policies, given that the quality of augmented data are beyond a reasonable level. We revisited AutoAugment series augmentation methods with a thorough policy search over the large search space defined by the permutation of operations with parameters, these methods apply AutoML search algorithms on augmentation for obtaining optimal augmentation policies for given datasets and models. The optimal policies implicitly means the ways of augmenting the most deficient and in turn the most effective data. Apparently these searched policies may improve the training efficiency in certain period of training, but from the view of whole training cycle, the searched results may not be optimal and worth the cost. We realized it could be possible to skip it for better efficiency without sacrificing the performance. Figure 4 explains the reason from an abstract view in the context of automated augmentation.

We describe the inherent logic of search-based and random augmentation from a general and abstract view of complementing deficient data, as shown in Figure 4. We think of the training data as a pool of knowledge or features with multiple dimensions. The columns in the histogram indicate the volume of data with this knowledge dimension. The black bars shows the data distribution of the original training data over the knowledge dimensions; while green, blue, yellow bars stand for data generated from the 1st to 3rd phases of augmentation. The red dotted lines show the data amount of the most deficient dimension upon completion of certain training phase with augmentation. With searched static policies of AutoAugment methods (as shown in Figure 4), the buckets of the most deficient data dimensions (e.g. dimension 2 in the Figure 4) can be filled fast once training starts and catch up with other dimensions in a relatively short period. In contrast, data growing with random policies are balanced amongst all dimensions in a larger sampling scope (Figure 4(b)), so data accumulation on the dimensions of deficient data may be slower than search-based methods. Hence, in early stage of training, a relatively slower accuracy increase is expected. We refer the time period required by stochastic approach to accumulate enough amount of data in the deficient dimensions as Stochastic Accumulation Stage (SAS). However, the relative deficiency are changing over time along training (as shown in Figure 4); with static policies searched by algorithms such as AutoAugment and Fast AA, the constant increasing on the focused dimensions may not fit the dynamic state of training and the data augmentation on the latest deficient dimensions slows down. Therefore, as training time goes, the data amount of deficient dimension in random approach may gradually get close to or even go beyond the one in search-based methods. When enough number of epochs is reached, the performance of random policy may match or overtake search-based methods. Our hypothesis is confirmed with experiment results on multiple datasets, as shown in Figure 3. As we see in the figure, the SAS periods of stochastic method are shorter than the total epochs spent on achieving optimal results in search-based methods, thus the slower accumulation of deficient data with random policies can be ignored while the benefit is it completely skips the expensive policy searching. We have to mention that succeeding research in Adversarial AutoAugment exploited adversarial network for generating dynamic policies which to some extent made up for the disadvantages of static policies. But the time and cost of policy searching were still existing and considerable, compared to the stochastic approach without any search.

Different from existing AutoAugment methods which focus on searching optimal policies, we tackle the problem from other views including introducing new augmentation operations and applying general augmentation strategy, which are presented in the following sections.

## A.2 HYPERPARAMETERS

In Table 7 we reported the details of the hyperparameters in our experimentens.

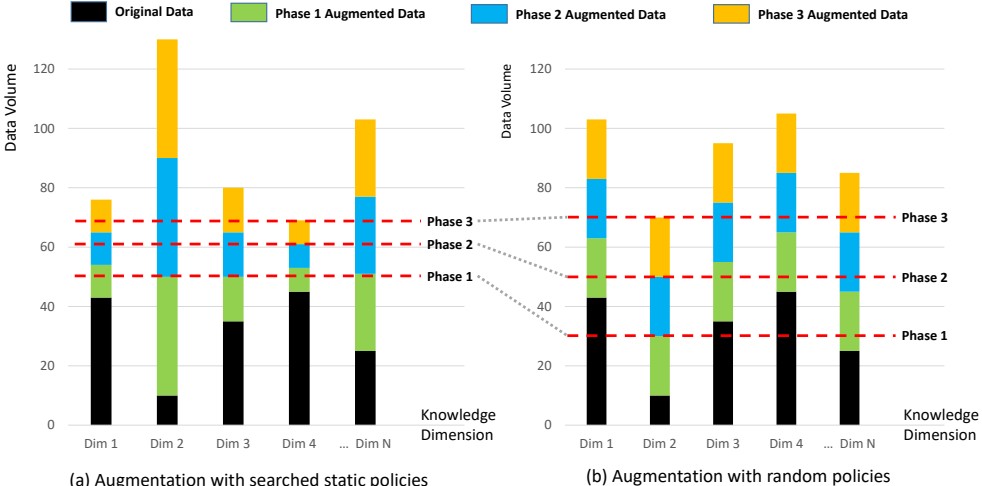

Figure 4: An abstract view of augmentation on deficient data. The black bars shows the data distribution of the original training data over the knowledge dimensions; while green, blue, yellow bars stand for data generated from the 1st to 3rd phases of augmentation. The red dotted lines show the data amount of the most deficient dimension upon completion of certain training phase with augmentation.

Table 7: Details of the hyperparameters in our experiments: We used the models: Wide-ResNet (40-2, 28-2, 28-10) (Zagoruyko & Komodakis, 2016), Shake-Shake (26 2x32d, 26 2x96d, 26 2x112d) (Gastaldi, 2017), PyramidNet+ShakeDrop (Han et al., 2017; Yamada et al., 2019) and ResNet-50 (He et al., 2016).

| Dataset | Model | Batchsize | initial LR | LR schedule | weight decay |
|---|---|---|---|---|---|
| CIFAR-10 | Wide-ResNet-40-2 | 256 | 0.1 | cosin | 2e-4 |
| | Wide-ResNet-28-10 | 256 | 0.1 | cosin | 5e-4 |
| | Shake-Shake(26 2x32d) | 256 | 0.01 | cosin | 1e-3 |
| | Shake-Shake(26 2x96d) | 256 | 0.01 | cosin | 1e-3 |
| | Shake-Shake(26 2x112d) | 256 | 0.01 | cosin | 2e-3 |
| | PyramidNet+ShakeDrop | 256 | 0.05 | cosin | 5e-5 |
| Reduced CIFAR-10 | Wide-ResNet-28-2 | 128 | 0.01 | cosin | 1e-3 |
| | Wide-ResNet-28-10 | 128 | 0.01 | cosin | 1e-3 |
| | Shake-Shake(26 2x96d) | 128 | 0.2 | cosin | 1e-4 |
| CIFAR-100 | Wide-ResNet-40-2 | 256 | 0.1 | cosin | 2e-4 |
| | Wide-ResNet-28-10 | 256 | 0.1 | cosin | 5e-4 |
| | Shake-Shake(26 2x96d) | 256 | 0.01 | cosin | 1e-3 |
| SVHN | Wide-ResNet-28-10 | 256 | 0.01 | cosin | 5e-4 |
| ImageNet | ResNet-50 | 512 | 0.05 | step LR | 1e-4 |

## A.3 STANDARD BENCHMARK TRAINING DETAIL

**CIFAR10/Reduced CIFAR10/CIFAR100/SVHN training details:** For CIFAR10, we perform experiments with the following models including Wide-ResNet-40-2, Wide-ResNet-28-10 (Zagoruyko & Komodakis, 2016), Shake-Shake(26 2x32d), Shake-Shake(26 2x96d), Shake-Shake(26 2x112d), and PyramidNet+ShakeDrop (Yamada et al., 2019). In addition, we evaluated our method on reduced CIFAR10 dataset which includes 4K training samples (randomly chosen) with Wide-ResNet-28-2, Wide-ResNet-28-10 and Shake-Shake(26 2x96d) models. We experiment with cifar-100 on Wide-ResNet-40-2, Wide-ResNet-28-10, Shake-Shake(26 2x96d), Shake-Shake(26 2x112d) and PyramidNet+ShakeDrop. These are the same models used in the experiments conducted in Fast AA. For SVHN dataset, we experiment with Wide-ResNet-28-10 on the core data of SVHN. Cosine learning rate scheduler and mini-batch size of 256 are adopted in all of our experiments.

As training epochs for various models are different, the complexity phases are defined respectively according to individual models. For Wide-ResNet-40-2 and Wide-ResNet-28-10, three phases are 200, 30, and 20 epochs respectively. For Shake-Shake(26 2x32d), Shake-Shake(26 2x96d), Shake-Shake(26 2x112d) and PyramidNet+ShakeDrop, three phases are 1800, 300, and 100 phases respectively. For reduced CIFAR10, three phases are 450, 30, and 20 epochs for Wide-ResNet-28-2, Wide-ResNet-28-10 models and 1500, 200, 100 epochs for Shake-Shake(26 2x96d) model.

ImageNet training details: For Imagenet, we conduct experiment with ResNet 50. The complexity phases are 270, 30, 20 epochs respectively. We use step learning scheduler in the experiment to keep consistent setting with Fast AA.

## A.4 EXPERIMENTS ON NUMBER OF OPERATION

We explore the influence of different number of operations in a sub-policy on stochastic augmentation in this experiment. As shown in figure 5, when number of operations in a sub-policy is larger than 3, the downgrade of performance is clear. Our interpretation is overlapped operations beyond certain number may cause downgrade of image quality and in turn affect the augmentation performance negatively.

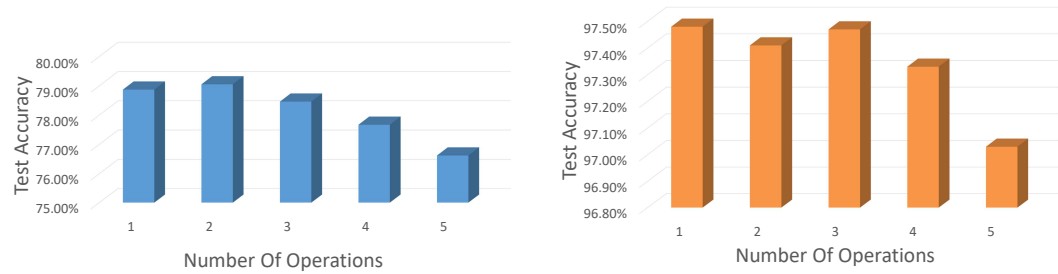

(a) Test Accuracy with WResNet-40-2 on CIFAR-100    (b) Test Accuracy with WResNet-28-10 on CIFAR-10

Figure 5: Influence of number of operations in a sub-policy on stochastic augmentation

## A.5 MULTI-STAGE TRAINING LOSS PROFILE

Figure 6 shows the training loss profile for different epoch allocation of multi-stage augmentation.

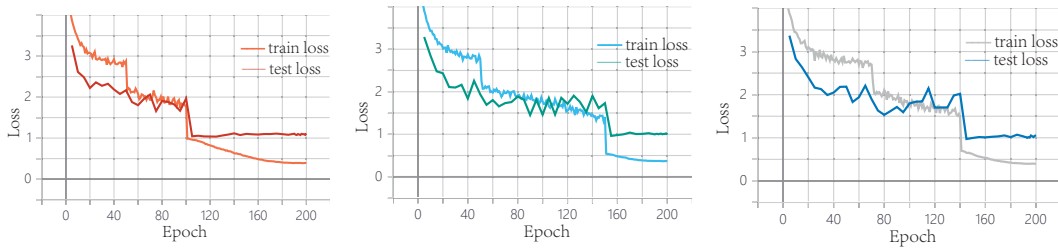

(a) Three stage strategy with 50, 50, 100 epochs allocated for stage 1, 2, 3 respectively

(b) Three stage strategy with 50, 100, 50 epochs allocated for stage 1, 2, 3 respectively

(c) Three stage strategy with 70, 70, 60 epochs allocated for stage 1, 2, 3 respectively

Figure 6: Training loss with various stage-epoch allocation strategy

## A.6 FACE RECOGNITION TRAINING DETAIL

We trained the system using MobileFaceNet (Chen et al., 2018) architecture with ArcFace loss (Deng et al., 2018). Training backbone is adapted to Fast AA system and the training conducted with the same settings described in Fast AA study apart from the search depth parameter which set as 100. We adapt the learning rate of 0.1 with 512 batch size to speed up the training. We trained the model for 50 epochs and learning rate is divided by 10 on epoch numbers 10,20 and 30.

3 stages training approach is done with 3 new additional augmentation methods. One of the additional operations is aging process which is based on StarGAN (Choi et al., 2017). In order to achieve aging effect, StarGAN is trained on Cross-Age Celebrity Dataset(CACD) (Chen et al., 2014). This dataset contains 160k images of 2000 celebrities with age ranges between 16 and 62. After aligning the images in the dataset, age ranges are set as 10-20, 20-30, 30-40, 40-50, 50-99 and StarGAN system is trained with these 5 classes by using default parameters. Subsequent to getting results, all classes are selected except 10-20 age range as the potential augmenters. Other additional operation is pose change and it is provided by employing (YadiraF). In order to achieve side faces, a given face image is turned on the x-axis with the angles defined between -45 and 45 degrees. The last of the additional operations is expression change. Expression change model is based on GAN-imation (Pumarola et al., 2018). For expression change augmentation, each image from CASIA is selected and augmented with 8 different (angry, disgusted, happy, sad, contemptous, fearful, neutral, suprised) expressions.

In the first stage of training all of the operations including the additional ones are used for 14 epochs. In the second stage, the additional complex augmenters turned off and training continued until 20th epoch. In the last step, all of the augmenters turned off and the whole training is completed.

## A.7 TEXT DETECTION TRAINING DETAIL

We trained our implementation of EAST (Zhou et al., 2017) architecture that uses PVANET (Kim et al., 2016) as its backbone with ICDAR 2017 MLT dataset. In addition, this model is integrated into Fast AA system and several experiments are performed with this system to provide comparative results. ICDAR 2017 MLT dataset provides 7.2K images for training, 1.8K images for validation, and 9K images for testing. The learning rate of 0.001 with cosine learning rate scheduler and mini-batch size of 24 are adopted, and the same settings are used for all experiments that we conducted for text detection task.

We also added 3 new geometry based (Distort, Stretch, Perspective) augmentations from (Luo et al., 2020) and a color based (LocalGamma) augmentation to the default augmentation list while keeping the baseline augmentations same with the original EAST (Zhou et al., 2017). These three geometry based augmentations are applied only to the bounding box areas. It is good to inform about that these additional augmentations are integrated to Fast AA system for comparing it with our method in equal conditions.

3 separate models are trained for our experiments in text detection task and we evaluated them over 9K test images with the standard evaluation method for the text localization task of ICDAR MLT 2017 competition.

We trained a model with only applying the baseline augmentations. Another model is trained with using the policies that are found by Fast AA system. For the last model that is trained by using our approach, we applied 2 stages augmentation along training. In the first stage, the baseline augmentations with 1.0 probability and the remaining augmentations including the additional ones with 0.5 probability are applied for 160 epochs and the baseline augmentations are applied only for the last 40 epochs.

It is also good to note that all models that we mentioned above are trained from the scratch without using any pre-trained weights for total of 200 epochs.

**E-Score: A fast and approximate metric for EAST RBOX text detector**

To speed up both the training and the evaluation phases, a customized metric is used in our experiments. This metric is faster to calculate and provides an approximate for F1 score. To give a further explanation, EAST is evaluated over a dataset by F1-score given a fixed IOU threshold like the other

text detectors. Though EAST, as its name suggests, is efficient during evaluation phase, but its still not efficient enough when embedded into training and validation processes. Therefore, our goal is to find a light but approximate enough method for EAST RBOX to estimate its final performance by F1 score. Our metric only works for EAST RBOX and it consists of two parts. The first part reflects how well the score map is learned and another part is for the RBOX geometry map. The algorithms to calculate the final metric given in the followings.

---

**Algorithm 1** The Algorithm to Evaluate the Score Map

---

**Input:** Real score map $\mathbf{Y}$, EAST RBOX's raw score map (logits) $\mathbf{L}$, the training mask $\mathbf{M}$, a pre-fixed thresh $\mathbf{T}$
**Output:** $f_1$, a floating value representing how well the score map learned.
 1: **Predict score map:**

$$\mathbf{Y}^* = sigmoid(\mathbf{L}) \tag{3}$$

 2: **Do masking:**

$$\hat{\mathbf{Y}}_{\mathrm{M}} = \mathbf{Y} \cdot \mathbf{M} \tag{4}$$

$$\mathbf{Y}_{\mathrm{M}}^* = (\mathbf{Y}^* > T) \cdot \mathbf{M} \tag{5}$$

 3: **Get Statistics:**

$$TP = \sum_{b,i,j} \hat{\mathbf{Y}}_{\boldsymbol{M}} \cdot \mathbf{Y}_{\boldsymbol{M}}^* \tag{6}$$

$$\hat{S} = \sum_{b,i,j} \hat{\mathbf{Y}}_{\boldsymbol{M}} \tag{7}$$

$$S^* = \sum_{b,i,j} \mathbf{Y}_{\boldsymbol{M}}^* \tag{8}$$

 4: **Get raw F1 score:**

$$f_1 = \frac{2\mathrm{TP}}{\hat{S} + S^* + \epsilon} \tag{9}$$

 5: **Return** $f_1$

---

---

**Algorithm 2** The Algorithm to Evaluate the Score Map

---

**Input:** the real RBOX map $\hat{G}$, the predicted RBOX map $G^*$, the positive sample mask $M_p$
**Output:** $g$, a floating value representing how well the RBOX map learned.
 1: **Retrieve AABB geometry and rotation angle:**

$$\hat{\mathbf{R}}, \hat{\mathbf{\Theta}} = split(\hat{\mathbf{G}}) \tag{10}$$

$$\mathbf{R}^*, \mathbf{\Theta}^* = split(\mathbf{G}^*) \tag{11}$$

 2: **Get point-wise IoU and cosine:**

$$\mathbf{IoU} = \frac{\left|\hat{\mathbf{R}} \cap \mathbf{R}^*\right|}{\left|\hat{\mathbf{R}} \cup \mathbf{R}^*\right|} \tag{12}$$

$$\boldsymbol{cosine} = ReLU(cos(\hat{\mathbf{\Theta}} - \mathbf{\Theta}^*)) \tag{13}$$

 3: **Get mixed value and do masking:**

$$g = \frac{1}{|\mathbf{M_p}|} \sum_{b,i,j} \mathbf{IoU} \cdot \boldsymbol{cosine} \cdot \mathbf{M_p} \tag{14}$$

 4: **Return** $g$

---

The final result of E-Score metric is shown in equation 15:

$$m = f_1 + g \tag{15}$$

There are several benefits of using E-Score, which are stated below:

- Calculation of $m$ is 'cheap', because it can be easily implemented and computed in the forward-pass. According to that statement, EAST RBOX's performance can be more traceable during both of the training and the validation phases. In addition, it is possible to integrate $m$ into Fast AA's search procedure to speed up overall policy search phases.

- Because both $f_1$ and $g$ are within $[0,1]$, $m$ is within $[0,2]$ and hence bounded. It is a good characteristic for value to be a metric.

- The real F1 score depends on many factors such as text score thresholds, implementations of non-maximum suppression (NMS) variants, and different datasets. As long as $m$ is positively correlated to the real F1 score, it is a close estimate to the real F1 score, and therefore, can be a good measurement of how well the model is trained.

