# OpenReview forum: "NOSE Augment: Fast and Effective Data Augmentation Without Searching"
_ICLR.cc/2021/Conference — Reject_

### Official Review · AnonReviewer3 · 2020-10-28
**Stochastic augmentation at three stages**

**Rating:** 5
**Confidence:** 4

**Review:**

The authors propose a method for learning an augmentation pipeline for image recognition. As opposed to recent existing approaches such as AutoAugment or RandAugment, the authors do not seek for the augmentation pipeline iteratively. Instead they use a stochastic approach, where augmenters are split to three categories based on their complexity to be used by curriculum learning.

The exact description of the selection method is not clear. Despite reading it several times, I did not fully understand the procedure. Moreover, the method and the split into the three categories (BaseAug, AdAug, SuperAug) seem somewhat arbitrary, and questions the validity of the results.

The results seem appropriate, and suggest that the proposed method can reach the state of the art; outperforming AutoAugment and Adversarial AutoAugment with a fraction of computational cost. However, from a practical standpoint, I consider all results in Table 1 almost equal. The minor differences in accuracy are not justifying the use of the proposed method. Instead, the authors should emphasize even stronger the computational savings; against all methods, not just two.

Finally, it is critically important that the authors specify what is the baseline (also called "standard augmentation").

---

> ### Author Response · Authors · 2020-11-17
> **Response to Review #3 (part 1)**
>
> We want to express our deep gratitude for your constructive suggestions and comments. Please find our replies as follows.
>
> Response to "exact description of the selection method" and "the method and the split into the three categories (BaseAug, AdAug, SuperAug) seem somewhat arbitrary"
> Thank you for your careful reading and constructive comments. We are sorry that our description of the augmenter selection method was not clear enough which might have caused some misunderstanding. We'll try clarifying here and update our paper accordingly. For the three augmentation phases, our method removes certain category of augmentations when it proceeds to the next augmentation phase. The first phase with the highest complexity level, includes all three main augmentation categories (CAT1: 15 transformation operations such as rotate, shear, translate etc.; CAT2: 3 mix-based operations including mixup, cutmix, augmix; CAT3: various baseline operations like flip, crop, and cutout depending on datasets); the second phase removes transformation operation category; and the third phase additionally removes mix-based operation category, with only baseline category left. As for the actual augmentation operations selected in the augmentation policy, the principle it follows is: when transformation category or mix-based category augmentations are applied, the actual operations are selected based on stochastic mechanism; while for baseline category the corresponding operations for specific datasets are fully applied during the whole augmentation process, which keeps consistent with the usage of baseline augmentations in related research work such as AutoAugment (Cubuk et al., 2019), RandAugment(Cubuk et al., 2020)., and Fast AutoAugment(Lim et al., 2019) etc. As we can see, the split of augmentations into the three categories (BaseAug, AdAug, SuperAug) is simply aligned with combinations of the three main augmentation categories without complex logics:
> -	BaseAug: baseline category;
> -	AdAug: baseline category + mix-based category;
> -	SuperAug: baseline category + mix-based category + transformation category.
> We hope the above description can help clarify our method and address your concerns.
>
> (To be continued with response part 2)

---

> ### Author Response · Authors · 2020-11-17
> **Response to Review #3 (part 2)**
>
> (Continued with part 1)
>
> Response to "computational savings":
>
> Thank you for your careful reading and constructive comments. The reason we didn’t include efficiency comparison with more methods in the first submission was that some of these methods didn’t describe their method efficiency in a very clear way for easy comparison. But enlightened with your constructive comment, we further analyzed the related works and added search cost estimations for Fast AutoAugment [2] and PBA [3] methods into Table-2.1 below. It can be seen that, For searching part, the cost of a search-based method is normally proportional to some factors such as model size, dataset size etc. For example, Fast AutoAugment's policy-search computation cost estimation is 3.5 GPU-hours for a small sized model and dataset (wresnet 40-2 with reduced cifar-10), time requirement for a larger model like Pyramid-Net+ShakeDrop is 780 GPU-hours on the same dataset.  Besides model and dataset sizes, the computation cost increases fast when number of the operations in augmentation pool increases. Similarly with other search-based methods, the search cost is generally very high. We measure searching cost in units of GPU hours. Table-2.1 below shows the searching costs of different augmentation methods. Here, the search cost of Adv AA was also claimed to be 0 as it does not have a separate policy search phase, but its training cost is much higher than our method, which is explained further in the following analysis of training cost.
>
> As for training part, NOSE augment has similar cost with the majority of search-based methods like AutoAugment and Fast AA, which process the training samples in one round. Suppose the training cost is a constant value “C”, which is affected by the dataset size and the number of total training epochs. For Adv AA, the training cost is at least 8 times C, because it augments more data in one epoch (number of batches is 8 times compared to methods such as NOSE, AutoAugment, and Fast AA). RandAugment [4] has significantly reduced the search space to 10x10, but it still relies on grid search with training and has a relatively higher training cost than NOSE augment.
>
> In general, our method has overwhelming advantage in overall efficiency as it bypasses policy search completely and does not increase training cost with any tuning or optimization logic.
>
> Table-2.1: Search space and search costs (GPU Hours)
>
> **Method**|**Search Space**|**R-Cifar10 (wresnet40)**|**Cifar10(pyramidnet)**|**R-ImageNet(wresnet40)**
> :-----:|:-----:|:-----:|:-----:|:-----:|
> AA|$10^{32}$|5000|-|1500
> FastAA|$10^{32}$|3.5|780|450
> PBA|$10^{61}$|5|-|-
> Our|$0$|$0$|$0$|$0$
>
>
> edit 1: we have updated our response with additional findings which have been already included in the new revision of the paper.
>
> (To be continued with part 3)

---

> ### Author Response · Authors · 2020-11-17
> **Response to Review #3 (part 3)**
>
> (Continued with part 2)
>
> References for response to "computational savings":
> [1] Ekin D Cubuk, Barret Zoph, Dandelion Mane, Vijay Vasudevan, and Quoc V Le. Autoaugment: Learning Augmentation policies from data. arXiv preprint arXiv:1805.09501, 2018.
> [2]  Sungbin Lim, Ildoo Kim, Taesup Kim, Chiheon Kim, and Sungwoong Kim. Fast autoaugment. arXiv preprint arXiv:1905.00397, 2019.
> [3]  Daniel Ho, Eric Liang, Ion Stoica, Pieter Abbeel, and Xi Chen. Population based augmentation: Efficient learning of augmentation policy schedules. arXiv preprint arXiv:1905.05393, 2019.
> [4]  E. D. Cubuk, Barret Zoph, Jonathon Shlens, and Quoc V. Le. Randaugment: Practical automated data augmentation with a reduced search space. 2020 IEEE/CVF Conference on Computer Vision and Pattern Recognition Workshops (CVPRW), pp. 3008–3017, 2020.
>
> Response to "specify what is the baseline":
>
> Thank you for your careful reading and reminder of the baseline specification. We are sorry we might overlook the baseline definition and missed including some reference in our first submission. We'll update our paper for a clearer baseline definition and as a quick clarification, we are using the same baseline as in related research work AutoAugment (Cubuk et al., 2019), and the successive research such as RandAugment(Cubuk et al., 2020) and Fast AutoAugment(Lim et al., 2019), as our paper aims to solve the same problem. Specifically, the baseline definitions in AutoAugment for different datasets are: for CIFAR-10 and CIFAR-100, standardizing the data, using horizontal flips with 50% probability, zero-padding and random crops, for SVHN, no special augmentations except standardizing the data; for ImageNet, standard Inception-style pre-processing which involves scaling pixel values to [-1,1], horizontal flips with 50% probability, random crops and random distortions of colors. We hope the above clarification helps address your question.
>
> (End of response to review 3)

---

### Official Review · AnonReviewer1 · 2020-10-28
**Review of NOSE Augment**

**Rating:** 3
**Confidence:** 5

**Review:**

This paper aims to provide an effective augmentation strategy without the need for a separate search. The resulting method is called NOSE Augment, which is presented as a substitute for the previous AutoAugment type methods (e.g. Fast AutoAugment, Population Based Augmentation, RandAugment, Adversarial AutoAugment etc.) In parallel to this goal, the authors propose adding the mixing-based augmentation operations Mixup, Cutmix, and Augmix into the list of operations used in AutoAugment. Finally, authors also employ a curriculum of augmentation strength during training.

While the cost of search for AutoAugment-type policies has been significantly reduced (e.g. PBA and Fast AA), it is still a worthwhile goal to want to remove the search phase completely, for further reduction of computational cost as well as convenience. While RandAugment does not use a separate search phase, it still has two hyperparameters that need to be optimized, similar to learning rate or weight decay, as the authors of this submission correctly point out. Thus I find the goal of wanting to remove the 2 hyperparameters that RandAugment requires worthwhile. However, I do not believe that the results shown in the paper indicate that NOSE Augment has achieved this goal, for reasons detailed below. Thus, I do not believe this work presents an improvement over previous work.

1)  RandAugment paper found that the optimal strength of augmentation depends on model size and dataset size, and found the optimal strength to be especially different for small ImageNet models such as Resnet-50 vs. large ImageNet models such as EfficientNet-B7. Since NOSE Augment is only evaluated on models that similarly sized to each other, it is hard for us to know if NOSE Augment would also do well on larger models such as EfficientNet-B7, or much smaller datasets that were explored in RandAugment. Looking at the optimal magnitude that was reported in RandAugment for the models NOSE Augment was evaluated on, it would not be surprising that random AutoAugment policies would do well. The authors should evaluate their method on a larger model such as EfficientNet-B7 or on a small dataset such as a small subset of CIFAR-10 to see the performance of NOSE Augment on different model sizes and dataset sizes.

2) This paper adds several different components to their method at once, without any ablations on where the improvements are coming from. Previous work has seen that combining AutoAugment and Mixup can be helpful (e.g. see  "Compounding the Performance Improvements of Assembled Techniques in a Convolutional Neural Network" by Jungkyu Lee et al.) In the case of NOSE Augment, it is not clear how much of their performance results from adding different methods on top of each other vs. other factors such as the curriculum.

3) It is also not clear to me what the main contribution of this paper is. Random AutoAugment policies have been evaluated both in AutoAugment and RandAugment papers. AutoAugment paper found that 25 random subpolicies did somewhere between Cutout and AutoAugment. RandAugment paper found that uniformly randomly sampled magnitudes do as well as constant magnitude on cifar-10. If the main contribution of NOSE Augment is the curriculum they use for augmentation, the authors should provide experiments that show that the curriculum alone gives improvements, and not the addition of new operations such as Mixup, Cutmix, and Augmix. PBA and RandAugment papers have both tried curriculum for augmentation, and it seems like RandAugment without curriculum can get as good results as PBA or RandAugment with curriculum.

4) For a paper on augmentation strategies, the results should only focus on improvements due to the augmentation strategy, and not any other model decisions such as training protocol or architecture. Looking at Table 1 of the paper, it is not clear to me if the authors have run their own baselines for each architecture, and what accuracy their own baselines got. The baseline column in Table 1 matches the baseline results that were reported in the AutoAugment paper to 4 significant figures for every row. This makes me think that the numbers reported as baseline are not actually the accuracies that the authors would necessarily get if they ran their experiments with just the baseline augmentation. For example, we know that ResNet-50 can get much higher accuracy than 76.3% on ImageNet, without any advanced augmentation, if other regularization methods are employed (e.g. label smoothing etc.) I would urge the authors to report the results of their own baseline models in all of the relevant tables, so that the reader can directly see how much of the reported performance is due to the proposed augmentation strategy.

---

> ### Author Response · Authors · 2020-11-17
> **Response to Review #1 (part 1)**
>
> We want to express our deep gratitude for your constructive suggestions and comments. Please find our replies as follows.
>
> Response to "if NOSE Augment would also do well on larger models such as EfficientNet-B7, or much smaller datasets that were explored in RandAugment"
>
> Following your kind suggestion, we have evaluated our method on reduced Cifar-10 dataset which includes 4K training samples. As one can see in the Table-1.1 (which will be also included in the new version of our paper) below, our method achieves better performance in terms of test accuracy (%) when you compare it with the reported values in RandAugment paper. Please also note that our re-produced results for the baseline experiments are consistent with the baseline results that reported in RandAugment paper.
>
> Table-1.1: Top-1 Test Accuracy (%) on Reduced-Cifar10
> +------------------------------+---------------+--------------------+-------+-------+----------------+
> |&nbsp;&nbsp;&nbsp;&nbsp;&nbsp;&nbsp;&nbsp;&nbsp;&nbsp;&nbsp;&nbsp;&nbsp;&nbsp;&nbsp;&nbsp;&nbsp;&nbsp;&nbsp;&nbsp;&nbsp;&nbsp;&nbsp;&nbsp;&nbsp;&nbsp;&nbsp;&nbsp;&nbsp;&nbsp;&nbsp;&nbsp;&nbsp;&nbsp;&nbsp;&nbsp;&nbsp;&nbsp;&nbsp;&nbsp;| baseline &nbsp;&nbsp;&nbsp;| baseline(our) | AA &nbsp;&nbsp;| RA &nbsp;&nbsp;| NOSE(our) |
> +------------------------------+---------------+--------------------+-------+-------+----------------+
> | Wide-ResNet-28-2 &nbsp;&nbsp;&nbsp;&nbsp;&nbsp;| 82 &nbsp;&nbsp;&nbsp;&nbsp;&nbsp;&nbsp;&nbsp;&nbsp;&nbsp;&nbsp;&nbsp;&nbsp;&nbsp;|  81.89 &nbsp;&nbsp;&nbsp;&nbsp;&nbsp;&nbsp;&nbsp;&nbsp;&nbsp;&nbsp;&nbsp;&nbsp;&nbsp;&nbsp;| 85.6 | 85.3 | 87.38 &nbsp;&nbsp;&nbsp;&nbsp;&nbsp;&nbsp;&nbsp;&nbsp;&nbsp;|
> +------------------------------+---------------+--------------------+-------+-------+----------------+
> | Wide-ResNet-28-10 &nbsp;&nbsp;&nbsp;| 83.5 &nbsp;&nbsp;&nbsp;&nbsp;&nbsp;&nbsp;&nbsp;&nbsp;&nbsp;&nbsp;| 83.16 &nbsp;&nbsp;&nbsp;&nbsp;&nbsp;&nbsp;&nbsp;&nbsp;&nbsp;&nbsp;&nbsp;&nbsp;&nbsp;&nbsp;&nbsp;| 87.7 | 86.8 | 89.18 &nbsp;&nbsp;&nbsp;&nbsp;&nbsp;&nbsp;&nbsp;&nbsp;&nbsp;|
> +------------------------------+---------------+--------------------+--------+------+----------------+
> |Shake(26 2x96d) &nbsp;&nbsp;&nbsp;&nbsp;&nbsp;&nbsp;&nbsp;&nbsp;&nbsp;| 82.9 &nbsp;&nbsp;&nbsp;&nbsp;&nbsp;&nbsp;&nbsp;&nbsp;&nbsp;&nbsp;| 80.26 &nbsp;&nbsp;&nbsp;&nbsp;&nbsp;&nbsp;&nbsp;&nbsp;&nbsp;&nbsp;&nbsp;&nbsp;&nbsp;&nbsp;&nbsp;| 89.98 | &nbsp;&nbsp;-&nbsp;&nbsp;&nbsp;| 89.98 &nbsp;&nbsp;&nbsp;&nbsp;&nbsp;&nbsp;&nbsp;|
> +------------------------------+---------------+--------------------+--------+------+----------------+
>
>
> For varying model sizes, Table 1.1 compares against the various AutoAugment series of works, including RandAugment. The table shows our proposed method performance to be on-par with RandAugment for various dataset sizes. The major difference is that our proposed work has no hyper-parameter to be tuned and hence no computational cost associated with policy search. By the way, we are also running experiments on much larger models such as ResNet-200 and EfficientNet-B7, but due to the large volume of computation resources required by these large models, we are trying our best but may not be able to complete executing these experiments by end of this rebuttal period. We will include these results into the paper when they are completed.
>
> Response to "ablations”, “it is not clear how much of their performance results from adding different methods on top of each other vs. other factors such as the curriculum”, and “provide experiments that show that the curriculum alone gives improvements”
>
> Thank you for your careful reading and comments. Here the overarching theme is that the whole is greater than the sum of its parts, i.e. the three components in our method need to work together to achieve a competitive performance with no policy search cost. The performance brought by individual components are evaluated in Section 4.8 of our paper with ablation experiments listed as follows:
> * “Stochastic + Base”: only transformation augmentations, no mix-based augmentations and phased augmentation. (Doesn’t work well)
> * “Stochastic + Two Stage”: phased augmentation (reversed curriculum) without mix-based augmentations (note that 2-stage but not 3-stage is used here because removing mix-based augmentations removes the corresponding phase, stochastic augmentations have to be kept otherwise there will be only one stage left). (Doesn’t work well)
> * “Stochastic+ mix-based”: mix-based augmentation when applied stochastically without phased augmentation. (Doesn’t work well)
> The conclusion of our ablation experiments (results shown in Table 6) is our proposed work is not incremental performance with each individual component, but rather, all three components need to work together to produce the strong performance we see in Table 1.
>
> (To be continued with response part 2)

---

> ### Author Response · Authors · 2020-11-17
> **Response to Review #1 (part 2)**
>
> (continued with part 1)
>
> Response to "the main contribution of this paper”:
>
> Thank you for sharing your views about the contributions of this paper. In our mind, the most significant of this paper is: by jointly applying phased (curriculum) augmentation and introducing more augmentation operations on top of simple stochastic augmentation mechanism, we find an effective and efficient method as an alternative of computation-intensive search-based augmentation methods. Not only is our proposed method a novel approach, our experiments proved the effectiveness and efficiency of our method, and the ablation experiments showed that all the components of our methods should be combined together to get better results. We hope our explanation here can address your concern.
>
> Response to "focus" of the paper and "baselines" of our experiments:
>
> Thank you for sharing your views of the focus of the paper. As a clarification, our work indeed focuses on augmentation strategy but not on any other model decisions such as training protocol or architecture, because we did not tune any hyper-parameters according to training protocol or architecture. We agree that AutoAugment (Cubuk et al., 2019) created a unique and interesting research branch which applies ideas and techniques of AutoML on augmentation strategies searching, and following this direction, a series of interesting research came up with focus on improving augmentation strategies. While our research tried tackling the same problem from a new point of view, in which we replaced search-based methods with an integral solution which jointly introduced new augmentation operations and phased augmentation strategy upon simple stochastic augmentation policies, and also obtained good performance in terms of both accuracy and efficiency.
> As our paper aims to solve the same problem, we continue to use exactly the same baseline which was used in AutoAugment, RandAugment and other comparison works. That is why you found the baseline results are the same as reported in the AutoAugment paper, and we are sorry if we didn't describe this relation clearly in our first submitted version. We are going to revise our paper accordingly and add proper reference of this baseline. 76.3% was exactly the baseline accuracy (Top-1) with ResNet-50 on ImageNet reported by AutoAugment (Cubuk et al., 2019), RandAugment(Cubuk et al., 2020)., and Fast AutoAugment(Lim et al., 2019); so we refer the same as the baseline in this experiment. We hope our explanation here can address your concern.
>
> (End of reponse to Review 1)

---

> > ### Comment · AnonReviewer1 · 2020-11-21
> > **thank you for the response and the additional experiments**
> >
> > Thank you for the response and the additional experiments. The results on reduced cifar-10 are very strong. I am excited to see the results on ResNet-200 and EfficientNet-B7 if they finish on time, but I understand that they may not.
> >
> > I am not sure your response addresses my concern about the baselines unfortunately. My concern is that you don't report your own no-augmentation baselines (e.g. just flips and crops for cifar-10, no augmentation for svhn etc.), but you copied the baseline numbers from tables in the AutoAugment paper. I think that it is crucial for augmentation papers to report their own baselines, so that the reader can see how much of the observed improvement is due to augmentations. I see that you have reported your own baselines for the recent reduced cifar-10 experiments, which is great. Is it possible to do this for the main results in table 1 (just the first column)?

---

> > > ### Author Response · Authors · 2020-11-22
> > > **about our no-augmentation (standard) baseline results**
> > >
> > > Thank you for your reply with inspiring words and kind understanding. Following your kind suggestion, we are collecting our no-augmentation (standard) baseline results. Based on the results we obtained so far, we can see that most of our baseline results are close to what AutoAugment reported in their paper, which validates that our baseline setting is consistent with AutoAugment and other related works. But in general, our baseline results are lower than AA's baseline more or less, it might be because we didn't spend time on tuning the hyperparameters at all. Our baseline results that we have obtained are listed as follows.
> > >
> > > **-**|**cifar10  base(our)**|**base**|**cifar100  base(our)**|**base**|**svhn base(our)**|**base**
> > > :-----:|:-----:|:-----:|:-----:|:-----:|:-----:|:-----:
> > > wr40-2|94.52|94.7|74.18|74|-|-
> > > wr28x10|95.43|96.1|80.79|81.2|96.57|96.9
> > > shake26 (2x32d)|95.78|96.4|-|-|-|-
> > > shake26 (2x96d)|96.65|97.1|79.77|82.9|-|-
> > > shake26 (2x112d)|96.68|97.2|-|-|-|-

---

### Official Review · AnonReviewer2 · 2020-10-29
**Official Blind Review #2**

**Rating:** 4
**Confidence:** 3

**Review:**

This paper proposes to completely remove even the two hyperparameters in RandAugment by full random selection of augmentation policies without any searching and achieve the comparable performances by simple multi-stage complexity driven augmentation strategy.

Pros.
It is interesting that the fully random selection of augmentation policies during training can produce competitive performances in comparison to the found augmentation policies by the previous augmentation search algorithms. In specific, with multi-stage training from hard augmentations to easy augmentations, it can improve the overall performances even when incorporating the mix-based augmentations which cannot be handled by the previous AutoAugment algorithms.
Experimental results on various tasks including not only image classification but also face recognition and text detection show that it consistently obtains improved performances over the baselines.

Cons.
It seems that the contribution of this work from the perspective of the algorithm novelty would be marginal from RandAugment. For example, the tuning for epoch allocation for each stage in the proposed method can be compromised with the hyperparameter search in RandAugment. And, the proposed manual three-stage augmentation scheduling can be compared with automatic augmentation scheduling such as PBA, AdvAA, and OHL-Auto-Aug (C. Lin, et al., Online Hyper-parameter Learning for Auto-Augmentation Strategy, ICCV 2019). Especially, compared with online augmentation learning in OHL-Auto-Aug that uses just 8 or 4 parallel trainings, the tuning for epoch allocation in the proposed multi-stage approach would not be computationally efficient, and it seems to largely affect the performances for each task, maybe network and dataset, etc.

Why the obtained results in Table 1 are different from Table 3 and Table 4 (97.97/80.27 vs. 97.75/79.89)?

It seems that the performance improvements by the use of mix-based augmentations are marginal.

---

> ### Author Response · Authors · 2020-11-17
> **Response to Review #2 (part 1)**
>
> We want to express our deep gratitude for your constructive suggestions and comments. Please find our replies as follows.
>
> Response to "algorithm novelty", " the tuning for epoch allocation", as well as "three-stage augmentation scheduling":
>
> We’d like to say our method is a new paradigm of augmentation method, which has essential difference with Rand Augment, AutoAugment, as well as other search-based methods. The novelty of our method is that it does not rely on any searching or hyper-parameter tuning but achieves better performance than computation-intensive search-based methods. Our method uses static complexity-driven strategy that are common for all datasets and models; this is distinct from existing automatic augmentation scheduling methods such as PBA, AdvAA, and OHL-Auto-Aug, which rely on either searching or hyper-parameter tuning with a cost of computation. While PBA, AdvAA etc. tunes policy hyperparameters (magnitude and probability) according to the training epochs, our three stage augmentation approach doesn’t change these hyperparameters but only reduces the augmentation categories from one augmentation stage to the next. This is corroborated by Table 2 (we have an updated Table 2.1 with more comparisons in the reply for reviewer3), where the computation overhead of a dynamic policy is contrasted clearly against our proposed static three stage policy e.g. Adversarial AA incurs 1280 hours of computation cost whereas our static policy requires a mere 190 hours.
> Specifically, our current method does not tune or search the optimal epoch allocation during the augmentation/training process; we have a common rule for epoch allocation for all three-phase experiments (except for the ablation study) with the 1st phase taking 80-85% of total epochs (which is mentioned in Section 4.5, 2nd paragraph), 2nd phase taking 10-13%, and 3rd phase 5-8%. We can see a small range here because we normally set epoch numbers round to 10 epochs (e.g. 200, 1800, 270) for simplicity purpose, it is not a result of tuning or searching, and this setting is not dynamically evaluated or updated. Theoretically, stage 1 is set to have the highest complexity and the majority (80-85%) of the epochs, because based on our study, it is necessary to maintain the high complexity and enough epochs to obtain good data diversity and prevent overfitting (details please refer to Section 3.3). Though this static phase/epoch allocation is decided partially based on empirical study, it is one-time setting for all cases and it is used as static rules. Our current experiment results are not guaranteed to be optimal in terms of epoch allocation as tuning/searching logic is not the focus of our method, no tuning or hyper-parameter searching is used in our method and this is the essential difference of our method in comparison with Rand or other search-based augmentation methods. However, your comment enlightened us that tuning or dynamic epoch allocation could be considered in future for optimizing our current method, given that we can find or create an efficient tuning/searching method.
>
> (To be continued with more responses for other comments)

---

> ### Author Response · Authors · 2020-11-17
> **Response to Review #2 (part 2)**
>
> (Continued with part 1)
>
> Response to "Why the obtained results in Table 1 are different from Table 3 and Table 4 (97.97/80.27 vs. 97.75/79.89)?":
>
> Thank you for your careful reading and question. The difference is because the settings are a bit different for these two experiments. CIFAR experiments in Table 1 are based on training of 250 epochs (three phases are 200, 30, and 20 epochs, this training setting is described in appendix A.4) for Wide-ResNet-40-2 and Wide-ResNet-28-10; while the ablation experiments in Table 3 and Table 4 are based on training of 200 epochs (as can be seen in Table 3(b), the sum of three stages for all cases are 200) to make setting consistent and results comparable with other ablation experiments. We are sorry that we didn’t make a very clear description of the setting, we will revise our paper with clearer setting description. But we hope our clarification can address your question.
>
> Response to "performance improvements by the use of mix-based augmentations are marginal":
>
> Thank you for your careful reading and comments. “Mix-based augmentations” alone is not our method, it must be applied in conjunction with the phased augmentation strategy in our method. There were actually some experiment details that we didn't elaborate much because of page limitation. In our experiment, we introduced multiple mix-based augmentations which may affect performance differently. There's also a chance that more than one mix-based operations can be applied on an image sample in one epoch. When these mix-based augmentations are introduced without phased augmentation strategy, it cannot bring significant performance improvements, and sometimes may even harm the performance (please refer to the experiment results shown in Table 6 and Table 1). But when mix-based operations are applied together with phased augmentation strategy, the negative results could be turned back to positive. And this is why our method combines sub-methods including stochastic policy, additional operations such as mix-based augmentation, and multi-stage strategy together and achieved apparent performance improvement as one integral solution. We consider this combination as one of the contributions of NOSE augment.
>
> (End of response to Review #2)

---

### Decision · Program_Chairs · 2021-01-07
**Final Decision**

**Decision:**

Reject

**Comment:**

All reviewers generally admit that the motivation of realizing search-free autoaugment is reasonable and important. However, they also raised many concerns regarding the experimental evaluation to validate the practical effectiveness of the method. In particular, unclear discussion with respect to ablation studies, and the lack of the baselines implemented by the authors were the central issues that obscure the essential effect of the core contribution of the work.  The authors made great efforts to conduct additional experiments and did address some of those issues, however some experiments are not yet ready such as the baseline implementation on ImageNet and testing on large models. After the discussion phase, all reviewers decided to keep their initial scores toward rejection, and the AC agreed with their opinions.

In summary, the paper focuses on an important problem and the proposed method is potentially very useful, but the paper in its current form should be further polished and completed before publication, thus I recommend rejection this time.